# REMIPS: Physically Consistent 3D Reconstruction of Multiple Interacting People under Weak Supervision

**Mihai Fieraru**[3]
mihai.fieraru@imar.ro

**Mihai Zanfir**[1]
mihaiz@google.com

**Teodor Alexandru Szente**[1,3]
teosz@google.com

**Eduard Gabriel Bazavan**[1]
egbazavan@google.com

**Vlad Olaru**[3]
vlad.olaru@imar.ro

**Cristian Sminchisescu**[1,2]
sminchisescu@google.com

[1]**Google Research** [2]**Lund University** [3]**Institute of Mathematics of the Romanian Academy**

## Abstract

The three-dimensional reconstruction of multiple interacting humans given a monocular image is crucial for the general task of scene understanding, as capturing the subtleties of interaction is often the very reason for taking a picture. Current 3D human reconstruction methods either treat each person independently, ignoring most of the context, or reconstruct people jointly, but cannot recover interactions correctly when people are in close proximity. In this work, we introduce **REMIPS**, a model for 3D Reconstruction of Multiple Interacting People under Weak Supervision. **REMIPS** can reconstruct a variable number of people directly from monocular images. At the core of our methodology stands a novel transformer network that combines unordered person tokens (one for each detected human) with positional-encoded tokens from image features patches. We introduce a novel unified model for self- and interpenetration-collisions based on a mesh approximation computed by applying decimation operators. We rely on self-supervised losses for flexibility and generalisation in-the-wild and incorporate self-contact and interaction-contact losses directly into the learning process. With **REMIPS**, we report state-of-the-art quantitative results on common benchmarks even in cases where no 3D supervision is used. Additionally, qualitative visual results show that our reconstructions are plausible in terms of pose and shape and coherent for challenging images, collected in-the-wild, where people are often interacting.

## 1 Introduction

Reconstructing three-dimensional models of multiple interacting people from images is an important computer vision task with applications in behavior analysis, automatic video analysis of sport events, or collaborative augmented reality applications. As the demands on the depth of analysis increase, beyond qualitative pose, one often seeks to know whether people are in contact or not, what is the nature of that contact, and how long it lasts. As such questions become important, it is clear that very coarse estimates of pose or even shape are no longer sufficient. One needs accurate models of shape and contact with predictable response over time. This is difficult due to the various degree of occlusion that occurs, the depth ambiguities particularly given only monocular images (but more generally given that occlusion and self-occlusion for people are frequent), the high degree of variability of even valid human poses, their different scales and underlying spatial image support, ranging from the large body parts like torso or thighs – informed by larger image regions – to hands, with parts typically accounted for by smaller spatial support, comparatively.

35th Conference on Neural Information Processing Systems (NeurIPS 2021).

Most of the volumetric human pose and shape recovery methods focus on a single person and require detecting the humans in the image and then running the inference model on each positive response. This approach has achieved good results and scales linearly in the number of people in the scene. There are multi people reconstruction methods which either predict 3D pose only and defer the shape reconstruction step to a later optimization step[1] or rely on an orthographic camera model [2]. However, it is nontrivial to obtain a realistic 3D scene placement for all the reconstructed human models such that their 3D spatial relations are plausible and the alignment with the image evidence is good. All aforementioned methods train using full 3D supervision coming from mocap datasets. However, this often lacks scenarios of interactions or fine grained self-contact. Recent work such as [3, 4] designed datasets for interaction analysis, with additional interaction signature annotations, and showed that models able to represent those produce more realistic reconstructions. However, they only work in an optimization framework and reconstruct only pairs of people in close interaction. Ideally one would like to be able to precisely model contacts, rely on attention models to identify the important element of interaction, have a framework that can accommodate more than two people and learn such models from data, using only weak supervision.

Our main contributions can be summarized as proposing the following: (1) fast, accurate and unified 3d self collision and interpenetration models for multiple people; (2) a novel vision transformer architecture to predict 3D pose and shape for multiple people; (3) weakly supervised models which do not require 3D annotations during training; (4) state of the art results on challenging datasets, with favorable performance compared to competing predictive or optimization-based methods.

In this paper, we propose **REMIPS**, a hybrid convolutional-transformer where the output of a human detector is combined with high-level convolutional image features. They are processed by a series of transformer encoder layers that iteratively refine the 3D pose and shape reconstruction estimates for all detected humans in the scene, irrespective of their number. Decoupling the detection component has the advantage of benefiting from innovations in that space without necessarily paying a performance cost (detection models are already quite fast, close to real time, and providing accurate results). Similar to [5], we use a full perspective camera model, allowing us to infer translations for all humans in the same camera coordinate system. We propose novel interpenetration and self-collision losses that are amenable to a deep-learning-based training process and work in a weakly-supervised regime.

## 2   Related Work

The field of 3D human pose and shape reconstruction from images has received increasing attention in recent years. This has been driven by progress in 3D statistical and articulated human body models [6, 7] and advances in 2D [8, 9, 10] and 3D [1, 11] pose estimation. Most reconstruction methods focus on estimating the 3D pose and shape of a single person [5, 12, 13, 14, 15] although, more recently, several papers focused on reconstructing multiple people coherently [2, 16, 1, 17].

Recent work started focusing on modeling inter-human contact [3] or self-contact [4, 18], developing methodology for more principled, contact aware reconstructions and releasing datasets with ground truth interaction [3] and self-contact [4] vertex-level annotations. We use these datasets in our training and evaluation pipeline, together with other various supervision signals.

Physically plausible human mesh reconstructions require collision avoidance for single and multiple interacting people. Prior work related to self-collisions [19, 20, 21, 22] rely on bounding volume hierarchies [23] to detect the set of colliding triangles. We propose novel, efficient and faster human mesh collision losses which are more appropriate for deep learning training pipelines. See §3.4 for more details.

Recent innovations in the space of visual transformers [24, 25] have increased the capabilities of neural network architectures for handling image features, and their ability to produce accurate reconstructions for images in-the-wild [26, 27]. We take inspiration from this as well.

## 3   Methodology

### 3.1   Statistical 3D Human Body Models

We use a recently introduced statistical articulated 3D human body model called GHUM [6], to represent the pose and the shape of the human body. The model has been trained end-to-end, in

a deep learning framework, using a large corpus of human shapes and motions. The model has generative body shape and facial expressions $\boldsymbol{\beta} = (\boldsymbol{\beta}_b, \boldsymbol{\beta}_f)$ represented using deep variational auto-encoders and generative pose $\boldsymbol{\theta} = (\boldsymbol{\theta}_b, \boldsymbol{\theta}_{lh}, \boldsymbol{\theta}_{rh})$ for the body, left and right hands, respectively, represented as normalizing flows [12]. The pelvis translation and rotation are controlled separately, and represented by a 6D rotation [28] $\mathbf{r} \in \mathbb{R}^{6 \times 1}$ and a translation vector $\mathbf{t} \in \mathbb{R}^{3 \times 1}$ w.r.t the origin $(0, 0, 0)$. The GHUM mesh, $\mathbf{M} = (\mathbf{V}, \mathbf{F})$, consists of a set of vertices $\mathbf{V} \in \mathbb{R}^{10168 \times 3}$ and a set of facets $\mathbf{F} \in \mathbb{R}^{20332 \times 3}$. In experiments we infer the GHUM parameters $(\boldsymbol{\theta}_b, \boldsymbol{\beta}_b, \mathbf{r}, \mathbf{t})$ omitting facial expression parameters $\boldsymbol{\beta}_f$ and the left and right hand pose parameters $\boldsymbol{\theta}_{lh}, \boldsymbol{\theta}_{rh}$, as we focus only on reconstructing the body pose and shape. We also drop the $b$ subscript for convenience.

## 3.2 Camera Model

We use a perspective projection operator $\Pi$ characterized by camera intrinsics $\mathbf{C} = [f_x, f_y, c_x, c_y]$ which assumes that the camera coordinate system is aligned with the world reference frame using an identity extrinsic rotation matrix and zero translation. This corresponds to a left handed coordinate system with positive $z$ values forward. All predictions are made in the camera coordinate system. When processing the camera intrinsics for a crop we use the same approach as in [5].

## 3.3 Architecture

In fig. 1 we show an overview of **REMIPS**, our proposed hybrid learning architecture for monocular multi-person 3D body pose and shape estimation. We draw inspiration from vision transformers [24], as we also use a hybrid convolutional-transformer architecture, and from [5], as we explore the idea of iteratively refining estimates by relying on cascaded, input-sensitive processing blocks, with homogeneous parameters, as an end-to-end learnable surrogate for non-linear optimization.

Our network receives as input an image $\mathbf{I} \in \mathbb{R}^{W \times H \times 3}$ containing multiple people, together with the pseudo ground-truth camera intrinsics $\mathbf{C} \in \mathbb{R}^{1 \times 4}$ of the image (see § 3.2). We apply a convolutional neural network (CNN) on the input image and extract a downsampled feature map representation $\mathbf{F} \in \mathbb{R}^{\frac{W}{32} \times \frac{H}{32} \times D}$. We flatten the feature map along the spatial dimensions to get a sequence of $N = \frac{W}{32} \times \frac{H}{32}$ tokens. This sequence is linearly embedded by means of matrix $\mathbf{E} \in \mathbb{R}^{D \times D'}$, where $D'$ is the embedding dimensionality. Next, learnable positional embeddings $\mathbf{E}_{pos} \in \mathbb{R}^{N \times D'}$ are added to the sequence. In addition to image feature tokens, we also use a sequence of $N_P = 16$ maximum tokens for the people in the image. We first run an off-the-shelf human detector on the input image and collect the bounding box coordinates for each person index $p$, $\mathbf{b}^p \in \mathbb{R}^{1 \times 4}$. For each person $p$, we compute its initial GHUM parameters $\mathbf{s}_0^p = \{\boldsymbol{\theta}_0, \boldsymbol{\beta}_0, \mathbf{r}_0, \mathbf{t}_0^p\}$ where we compute the translation such that the default GHUM mesh projects in the center of the corresponding bounding box $\mathbf{b}^p$. We denote by $\mathbf{S}_0 = [\mathbf{s}_0^1, \ldots \mathbf{s}_0^{N_P}]$ the initial state of GHUM parameters for all the people in the image. For each person, we concatenate the bounding person coordinates, the initial GHUM state and the camera intrinsics and construct the sequence of tokens corresponding to the persons in the image $\mathbf{P} \in \mathbb{R}^{N_P \times D_P}$. We linearly project the people token sequence $\mathbf{P}$ by means of matrix $\mathbf{E}_P \in \mathbb{R}^{D_P \times D'}$. The two sequences of tokens are concatenated into a single input sequence $\mathbf{Z}_0$ and iteratively transformed through a single shared transformer encoder layer [25], *TL*, for a number of $L$ steps. We collect at each step $l \in \{1 \ldots L\}$ a refinement update $\Delta \mathbf{S}_l$ from each transformed representation $\mathbf{Z}_l$, using a shared MLP applied on the corresponding transformed representation of the people token sequence.

$$\mathbf{Z}_0 = [\mathbf{F}_s \mathbf{E} + \mathbf{E}_{pos}, \mathbf{P} \mathbf{E}_P] \tag{1}$$

$$\mathbf{Z}_l = TL(\mathbf{Z}_{l-1}) \tag{2}$$

$$\Delta \mathbf{S}_l = MLP(\mathbf{Z}_l^{N:N+N_P}). \tag{3}$$

The refinement updates $\Delta \mathbf{S}_l$ are added to the initial state GHUM parameters, $\mathbf{S}_0$, as

$$\mathbf{S}_L = \mathbf{S}_0 + \lambda \Sigma_{l=1}^L \Delta \mathbf{S}_l \tag{4}$$

where $\mathbf{S}_L = \{\boldsymbol{\Theta}_L, \mathbf{B}_L, \mathbf{R}_L, \mathbf{T}_L\}$ are the final GHUM parameter estimates for all the people detected in the image and $\lambda$ is a weighting term for the residual estimates.

Note that the maximum number of person tokens $N_P = 16$ is chosen specifically for training purposes, to batch training examples together. Yet, at inference time, we can actually use as many

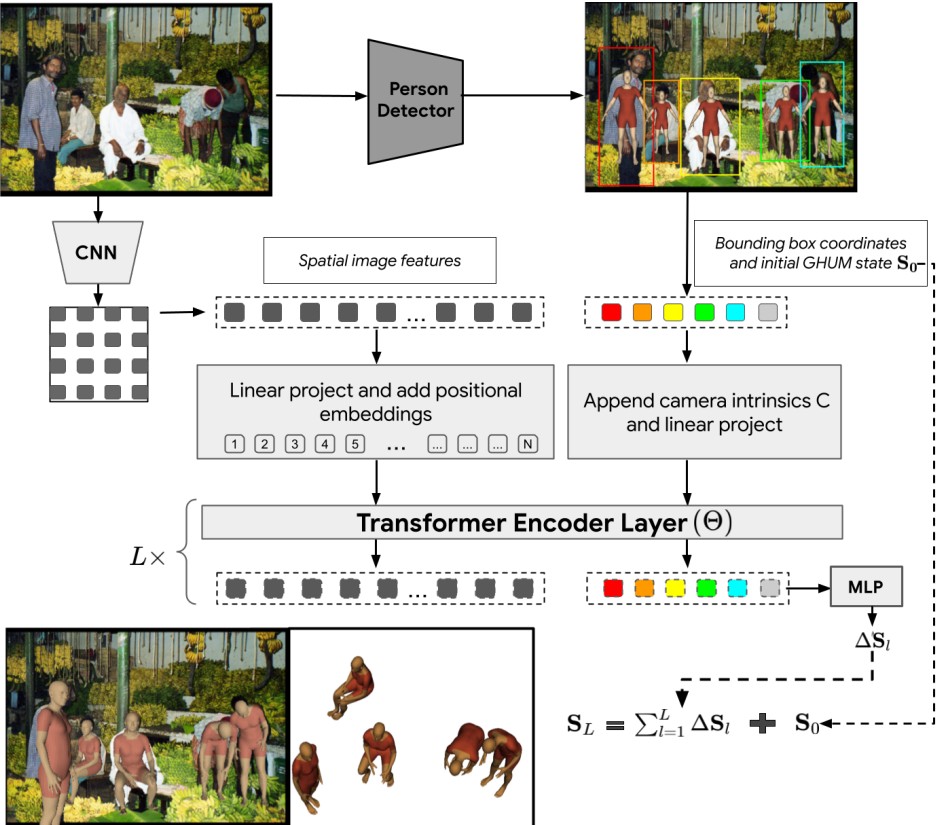

Figure 1: Overview of **REMIPS**, our proposed architecture to reconstruct the 3D pose and shape of multiple interacting people. Starting from a single input image, we use an off-the-shelf detector to extract the human bounding boxes. We create a sequence of person tokens from these detections to which we attach an initial GHUM state estimate $\mathbf{S}_0$. On a separate branch, starting from the image, we run a backbone convolutional neural network architecture and create an additional sequence of spatial image feature tokens, $\mathbf{F}_s$. We concatenate the two sequence representations and iteratively refine this joint representation through a single transformer encoder layer for a number of $L$ stages. At the end of each stage $l$, we collect the transformed representation for the token sequence associated with the people and apply an MLP to regress the residual GHUM state estimates $\Delta\mathbf{S}_l$. Our final estimation is given by the weighted sum of all the residual state updates and the initial state. The network is trained weakly-supervised on various datasets with 2D annotations. We use contact and collision losses defined over the recovered geometries to ensure physical plausibility.

person tokens as the person detector outputs (even more than 16), as long as the quadratic memory requirements for the attention mechanism are satisfied (i.e. in the order of thousands of persons). We observe that, in practice, the differences in inference speed for a variable number of persons (between 1 and 32) to be almost negligible, i.e. at most 10% increase in running time.

We train the network by optimizing the following loss function:

$$\mathcal{L} = \mathcal{L}_{sc} + \mathcal{L}_{ic} + \mathcal{L}_{ics} + \mathcal{L}_{scs} + \mathcal{L}_d + \mathcal{L}_{ka} + \mathcal{L}_\theta + \mathcal{L}_\beta \tag{5}$$

where the first 5 loss terms correspond to the physical plausibility of the reconstructed meshes as follows: $\mathcal{L}_{sc}$ and $\mathcal{L}_{ic}$ are the self-collision loss and the interpenetration loss, respectively; $\mathcal{L}_{ics}$ and $\mathcal{L}_{scs}$ correspond to the supervised self-contact [4] and interaction-contact [3] losses; $\mathcal{L}_d$ is the depth-aware ordering loss from [2]. These losses are discussed and explained in more detail in subsequent sections.

We use a standard **keypoint alignment loss**, $\mathcal{L}_{ka}$, that measures the error between the 3D joints from the GHUM reconstructions, projected onto the image plane, $\mathbf{J} = \{\mathbf{J}_i\}_{i=1,\dots,K}$, and the available 2D

keypoints annotations in the image $\{\mathbf{j}_i\}_{i=1,\ldots,K}$

$$\mathcal{L}_{ka} = \frac{1}{K} \sum_{i=1}^{K} \|\mathbf{j}_i - \Pi(\mathbf{J}_i, \mathbf{C})\|_2^2.$$  (6)

Note that for some datasets we do not have 2D pose annotations available. In that case we supervise with the predictions of an off-the-shelf 2D pose estimator.

For **body shape and pose**, we use regularization losses:

$$\mathcal{L}_\theta = \|\boldsymbol{\theta}\|_2^2, \quad \mathcal{L}_\beta = \|\boldsymbol{\beta}\|_2^2.$$  (7)

## 3.4 Physical Collision for GHUM

In our methodology, we use an approximation for the GHUM mesh topology, which we denote by $\hat{\mathbf{M}} = (\hat{\mathbf{V}}, \hat{\mathbf{F}})$. This mesh topology is obtained by applying decimation operators on the default GHUM model resulting in a mesh with $N_{\hat{v}} = 512$ vertices and $N_{\hat{t}} = 1020$ triangles. We refer to this low resolution approximation of the GHUM mesh as **PHUM**. As $\hat{\mathbf{V}}$ is a subset of the GHUM vertices, this easily allows the PHUM mesh to be reposed and reshaped with respect to the underlying GHUM mesh it approximates. The ICP loss of our approximation (*i.e.* for each vertex on the GHUM mesh not included in PHUM, we compute the distance to the closest point on the PHUM surface) is of 2.8mm. By using PHUM, we drastically reduce the computation cost of our proposed self-collision and interpenetration losses as they are quadratic in the number of vertices. We use generalized winding numbers [29] to rapidly test whether 3D points are inside/outside relative to a closed mesh topology.

**Self-collision loss**  Given a GHUM mesh, $\mathbf{M}$, we compute its PHUM approximation $\hat{\mathbf{M}}$. We apply the generalized winding number test $L = \phi_{\hat{\mathbf{M}}}(\hat{\mathbf{V}})$ on its set of vertices $\hat{\mathbf{V}}$, with respect to its own mesh topology $\hat{\mathbf{M}}$ and gather the binary inside/outside labeling $L \in \{0,1\}^{N_{\hat{v}} \times 1}$. The vertices marked as inside, $\hat{\mathbf{V}}_{L_+}$, with $L_+ = \{l \in L : l = 1\}$, are pushed out of the mesh in the direction of their nearest neighbor vertices, $\mathrm{NN}(\hat{\mathbf{V}}_{L_+}, \hat{\mathbf{M}})$:

$$\mathcal{L}_{sc} = \sum_{l \in L_+} \|\hat{\mathbf{V}}_l - \mathrm{NN}(\hat{\mathbf{V}}_l, \hat{\mathbf{M}})\|_2$$  (8)

The nearest neighbors are extruded outside the mesh by a small amount (*i.e.* 5mm) in the direction of their normal before computing the loss, such that inside vertices get pushed outside the mesh. For the GHUM mesh, we have body part labeling information available which is transferred onto PHUM. That is, for each vertex in the mesh we have its associated body part label class from a total of 14 body parts. When computing a nearest neighbor vertex we either consider vertices that are exterior or on another body part than the query vertex, and we choose the one with the smallest distance. The body part labeling allows us to avoid situations where an interior vertex can choose to exit the mesh geometry through a vertex from the same body part as itself.

**Interpenetration loss**  Interpenetration is straightforward to handle by our representation. Given a pair of meshes $\hat{\mathbf{M}}_1$ and $\hat{\mathbf{M}}_2$, we compute the generalized winding number test for both $L_1 = \phi_{\hat{\mathbf{M}}_2}(\hat{\mathbf{V}}_1)$ and $L_2 = \phi_{\hat{\mathbf{M}}_1}(\hat{\mathbf{V}}_2)$. The loss is then given by:

$$\mathcal{L}_{ic} = \sum_{l \in L_1+} \|\hat{\mathbf{V}}_{1,l} - \mathrm{NN}(\hat{\mathbf{V}}_{1,l}, \hat{\mathbf{M}}_2)\|_2 + \sum_{l \in L_2+} \|\hat{\mathbf{V}}_{2,l} - \mathrm{NN}(\hat{\mathbf{V}}_{2,l}, \hat{\mathbf{M}}_1)\|_2$$  (9)

This time, when we compute the nearest neighbor for inside vertices of a mesh, we query from all vertices in the colliding mesh.

**Relation to Prior Work**  Whereas in our formulation self-collision and interpenetration collision are handled based on a common unified representation, in the literature different approaches are used for each type of collision. For self-collision, most related work [19, 20, 21, 22] uses bounding volume hierarchies (BVH) [23] to detect a list of colliding triangles. Using local conic 3D distance fields, penetrations are penalized by the depth of the intrusion. The choice of penalizing collisions only for colliding triangles (at the border) is done solely for performance reasons (as pointed out in Section

3.2.4 from [20]). Ideally, one would use the full set of points inside the mesh and their distance to the surface, which is our case. We compare the running time of the forward pass of the self-collision loss proposed in [21] with that of our self-collision loss, based on PHUM and generalized winding numbers. For a batch size of 1, on one NVIDIA Tesla V100 32GB, our self-collision loss computation takes 33.5ms, while the one proposed in [21] takes 50.8ms. Differently from the BVH implementation, we did not write any custom CUDA kernel, but instead rely on standard TensorFlow [30] operations with batch support.

For interpenetration collision, the most popular approach [22, 2] is to first voxelize the two meshes and then compute a signed distance field (SDF) for each of them. The SDF of each mesh is queried by the vertices of the other mesh to compute the loss. The voxelization step takes around 45ms for a mesh and the SDF computation is implemented using a custom CUDA kernel. The running time is more than an order of magnitude higher than our generalized winding number computation and has a considerably more complex implementation.

### 3.5 Depth-Ordering Loss

We follow the same procedure as in [2] to train using a depth-ordering loss. We use ground-truth instance segmentation masks from COCO [31] during training. For our differentiable mesh rasterizer we use DIRT [32] to render the depth maps, $D_k$, for each person. We use the depth ordinal loss:

$$\mathcal{L}_d = \sum_{p \in S} \log(1 + \exp(D_{y(p)}(p) - D_{\hat{y}(p)}(p)) \tag{10}$$

where $S = \{p \in I : y(p) > 0, \hat{y}(p) > 0, y(p) \neq \hat{y}(p)\}$, $I$ is the RGB image, $y(p)$ is the index of the person occupying pixel $p$ and $\hat{y}(p)$ is the ground-truth index of the person occupying pixel $p$.

### 3.6 Contact Losses

Similarly to [3, 4], we employ contact losses when ground truth annotations are available. For a pair of people, $P_1, P_2$, in contact in an image $I$, we define the contact signature $C_{ics}(I, P_1, P_2)$ at the facet level [3] as a matrix where $C_{ics}^{f_1,f_2}(I, P_1, P_2) = 1$ if facet $f_1 \in \mathbf{M}(P_1)$ is in contact with facet $f_2 \in \mathbf{M}(P_2)$ and 0 otherwise, where $\mathbf{M}(P)$ is the mesh corresponding to person $P$. The distance between two facets $f_1$ and $f_2$ is defined as the Euclidean distance $d(f_1, f_2)$ between the centers of the two facets [3]. We define the following interaction-contact signature loss

$$\mathcal{L}_{ics}(I, P_1, P_2) = \sum_{f_1 \in \mathbf{M}(P_1), f_2 \in \mathbf{M}(P_2)} C_{ics}(I, P_1, P_2) d(f_1, f_2) \tag{11}$$

In a similar fashion, the self contact signature $C_{scp}(I, P)$ [4] of a person $P$ in an image $I$ is defined as $C_{scp}(I, P) = C_{ics}(I, P, P)$, where $C_{scp}^{f_1,f_2}(I, P) = 1$ if facet $f_1 \in \mathbf{M}(P)$ is in contact with facet $f_2 \in \mathbf{M}(P)$. Subsequently, the self-contact signature loss is defined as

$$\mathcal{L}_{scs}(I, P) = \sum_{f_1 \in \mathbf{M}(P), f_2 \in \mathbf{M}(P)} C_{scp}(I, P) d(f_1, f_2) \tag{12}$$

In practice, contact signature matrices are sparse (have less than 100 non zero values) so the computation of $\mathcal{L}_{ics}$ and $\mathcal{L}_{scp}$ while training our models is efficient.

## 4 Experiments

In this section, we offer implementation details regarding the network architectures and training hyperparameters, we describe the datasets used for training and evaluating our models, and we present state-of-the-art results on the challenging **Panoptic** [33] and **CHI3D** [3] datasets, as well as several studies illustrating the importance of the proposed components and design choices.

### 4.1 Implementation details

For all our models we use a ResNet50 [34] backbone pretrained on ImageNet [35]. The network has a total number of 26M parameters, out of which 23M parameters for the backbone and 3M parameters

for the transformer layers. We use $L = 4$ stages, an embedding size of 512 and 8 heads for the MultiHeadAttention layer. The network is trained for 150 epochs with a batch size of 32, starting learning rate of $1e - 4$ and exponential decay of $0.98$. Our code is implemented in TensorFlow and we employ random scaling, rotation, horizontal flipping and cropping augmentations during training. All images are resized to $480 \times 480$ while preserving the original aspect ratio using padding when necessary. All networks are trained on a single V100 GPU with 32GB of memory.

## 4.2 Datasets

In our experiments, we train weakly supervised on various datasets with 2D supervision in the form of annotated body joints and instance segmentation masks. When available, we use vertex-level self-contact and person-to-person contact supervision. Each dataset usually contains only a subset of annotation types, so we employ the corresponding losses accordingly.

**MPII** [36] is an indoor/outdoor image dataset of 25K images containing multiple people with around 40K 2D pose annotations. We only use a subset of 10k images of this dataset for training.

**PoseTrack** [37] is an in-the-wild dataset containing 1.3K video sequences with 2D poses for multiple people annotated across multiple frames. We use a subset of 25K images and 160K 2D pose annotations from the trainval subset of this dataset, for training only.

**LSP** [38] and **LSP Extended** [39] are in-the-wild datasets with single person 2D pose annotations. We use the training subsets consisting of approximately 1K, respectively 10K images with ground truth 2D pose annotations.

**COCO** [31] is a dataset containing 40K images with multiple people. In total, 100K annotated poses are used for training our models. Similarly to [2], the instance person segmentation masks are used for computing the depth-ordering loss, in training only.

**FlickrCI3D** [3] and **FlickrSC3D** [4] contain in-the-wild images with multiple people engaged in various activities involving human interactions and self-contacts. Ground truth interactions and self-contact annotations are available for this dataset at facet granularity. These datasets have no annotations for 2D keypoints, so we generate them by running a 2D human pose predictor on each of the images to obtain pseudo ground-truth 2D pose annotations. We use a total of 15K images with more than 25K interactions or self-contact annotations for training.

The **Panoptic** studio [33] is an indoor multiple people dataset captured in a multi-view laboratory system. We use the same data selection as in [16, 2] resulting in 9.6K images containing ground truth 3D pose for multiple people in each image. We use this dataset for evaluation purposes only.

**CHI3D** [3] is an indoor dataset with 3D ground truth joints from mocap containing images from 631 sequences and 4 cameras. Each sequence contains 2 people in various interactions and contact ground-truth annotations exist for each image. We use this dataset for evaluation purposes only.

**Human3.6M** [40] is a single-person 3d pose dataset containing sequences of people performing various activities, containing more than 3 million 3d skeletons. We use it only in one experiment (see table 1) for evaluation under protocol P1 and use no 3d ground truth during training.

## 4.3 Results

Although our method is designed to reconstruct any number of people, we first evaluate REMIPS on the popular **Human3.6M** dataset [40] to ensure it is competitive in the simpler single-person scenario. For this experiment only, REMIPS is fine-tuned on the training images from the P1 protocol of Human3.6m. We do this only in a self-supervised regime, i.e. fine-tuning only with supervision from predicted 2d keypoints, no ground truth 3d information being used. We compare favorably with [13] and [5] (table 1), which also report results of their methods trained without full 3d supervision.

We now start the multi-person evaluation of our model on the challenging **Panoptic** [33] dataset. We follow the evaluation protocol from [16, 1, 2] (without training on any data from the **Panoptic** dataset). We present state-of-the-art results in table 2, showing that **REMIPS** performs better than other optimization- [16] or inference-based multiple people reconstruction methods [1, 2].

We continue our evaluation on the **CHI3D** [3] dataset, which is not used during training. CHI3D consists of sequences of two interacting people, together with ground-truth motion capture markers,

| Method | MPJPE-PA | MPJPE |
|---|---|---|
| HMR [13] | 67.4 | 106.8 |
| HUND [5] | 66.0 | **91.8** |
| **REMIPS (ours)** | **64.3** | 96.1 |

Table 1: Evaluation on the **Human3.6M** dataset [40] - protocol P1. We report mean per joint positional error (**MPJPE**) and its version including Procrustes Analysis (**MPJPE-PA**). All errors are reported in mm.

| Method | Haggling | Mafia | Ultimate | Pizza | Mean |
|---|---|---|---|---|---|
| Zanfir *et al.* [16] | 140.0 | 165.9 | 150.7 | 156.0 | 153.4 |
| Zanfir *et al.* [1] | 141.4 | 152.3 | 145.0 | 162.5 | 150.3 |
| Jiang *et al.* [2] | 129.6 | **133.5** | 153.0 | 156.7 | 143.2 |
| **REMIPS (ours)** | **121.6** | 137.1 | **146.4** | 148.0 | **138.3** |

Table 2: Performance on the **Panoptic** [33] dataset for pose and shape reconstruction methods focusing on multiple people. All errors represent mean per joint position errors and are reported in mm, relative to the root joint. The evaluation protocol is the same as the one in [16, 1, 2]. Our model achieves lower error compared to optimization- or learning-based approaches.

3d joints and GHUM parameters available for one of the two. For each sequence, one contact frame is annotated with an interaction contact signature. We use the same evaluation protocol as in Fieraru *et al.* [3] and, for the contact frames, we compute the 3D joint error for the prediction which has the highest 2D bounding box overlap with the ground truth. In table 3, we present state-of-the-art results when we compare again against optimization [3] and inference only [2] methods. To run the model from [2] on the **CHI3D** [3] dataset, we use their publicly available code repository.

| Method | MPJPE | MPVPE | Translation Error | #2D | #3D |
|---|---|---|---|---|---|
| Fieraru *et al.* [3] | 125.4 | – | 368.0 | N/A | N/A |
| Jiang *et al.* [2] | 136.0 | N/A | N/A | 100K | 300K |
| **REMIPS (ours)** | **120.8** | **134.7** | **284.1** | 115K | 0 |

Table 3: Performance on the **CHI3D** [3] dataset for multiple person pose and shape reconstruction methods. In columns 2, 3, 4 we show the mean per joint position error (**MPJPE**), the mean per vertex position error (**MPVPE**) and the translation error. All errors are reported in mm and are relative to the root joint. Our method has lower errors compared to the other optimization and inference based methods. We also compare the number of **#2D** and **#3D** annotations used as supervision during the training of the different models. Our models use no **#3D** and achieve better performance on the challenging dataset **CHI3D** [3].

In table 4, we present a series of ablations on the **CHI3D** [3] dataset for the loss components in eq. 5 that are used during training. We also show the results for a version of REMIPS that does not take pseudo ground-truth camera intrinsics parameters as input. The lowest error is achieved using all of the introduced losses and inputting camera intrinsics, which supports their individual importance.

| Method | MPJPE | MPVPE |
|---|---|---|
| REMIPS (w/o $\mathcal{L}_{ic}$) | 122.7 | 134.9 |
| REMIPS (w/o $\mathcal{L}_{sc}$) | 123.5 | 137.1 |
| REMIPS (w/o $\mathcal{L}_{ics}$) | 127.1 | 142.1 |
| REMIPS (w/o $\mathcal{L}_{scs}$) | 125.3 | 140.0 |
| REMIPS (w/o camera instrinsics) | 139.0 | 158.0 |
| **REMIPS (ours)** | **120.8** | **134.7** |

Table 4: Ablation study on the **CHI3D** dataset for multiple person pose and shape reconstruction methods. We report mean per joint positional error (**MPJPE**) and mean per vertex positional error (**MPVPE**). All errors are reported in mm and are relative to the root joint. The model trained with all proposed losses $\mathcal{L}_{ic}, \mathcal{L}_{sc}, \mathcal{L}_{ics}, \mathcal{L}_{scs}$ and taking camera intrinsics as input achieves the lowest error.

To study how the number of people in the scene and the spatial context affect the performance of REMIPS, we run our method in three different ways and introduce a new baseline based on the single person body and pose estimator SPIN [14]. We evaluate each scenario on the **Panoptic** dataset [33] where we report the MPJPE under the same protocol as in [16, 1, 2], averaged across all frames and actions. We propose the following experiments: (1) *multi-person setting* - where the input image is a crop around all detected persons and bounding boxes are provided to the transformer together

for all persons in the crop (this is the original version of REMIPS); (2) *single-person setting with context* - where the input image is a crop around all detected persons and each bounding box is provided separately to REMIPS (i.e. one bounding box token at a time); (3) *single-person setting without context* - where each input image consists of a crop of only one detected person and REMIPS receives only its bounding box (i.e. like in a standard single-person 3d pose reconstruction setting, such as in SPIN [14], HMR [13], HUND [5]); (4) *single-person setting without context* (baseline with SPIN [14]) - where each input image consists of a crop of only one detected person that, this time, is processed by SPIN [14]. Results are illustrated in table 5. Comparing scenarios (1) and (2),

| Method | MPJPE |
|---|---|
| (1) multi-person (REMIPS) | 138.3 |
| (2) single-person w. context (REMIPS) | 138.2 |
| (3) single-person w/o context (REMIPS) | 141.7 |
| (4) single-person w/o context (SPIN) | 164.9 |

Table 5: Performance comparison on the **Panoptic** [33] dataset for different scenarios of REMIPS and a SPIN baseline. Scenarios (1) and (2) suggest that pose accuracy is not affected by the number of people detections. Comparing (2) and (3), we observe the benefits of providing the large context of the image as input, while comparing (3) with (4) highlights the accuracy of REMIPS over recent methods, even in the tight crop single-person scenario.

results are almost identical in terms of pose accuracy, leading us to suspect that, at inference time, attention to the image feature tokens is the most important part of the transformer architecture, given a larger context. This means that spatial image features already encode information regarding the other persons in the scene, due to the way the network is trained (i.e. reasoning about all the persons at once). Pose accuracy is not affected by the number of people in the input as long as the context is large/informative enough. Comparing scenarios (2) and (3), we see the benefit of providing the large context of the scene as input (3.5mm accuracy improvement). Comparing scenarios (3) and (4), we observe that even in a single-person scenario with crops tight around each person, REMIPS performs much better than SPIN (a significant difference of 23.2mm). Note that neither of the two methods was trained on Panoptic, and our method has not used any 3d ground-truth mesh supervision. We observe large errors for SPIN in cases of occlusions and close people interactions.

To study the effect that the distance between a person and the camera has on REMIPS, we perform another study on the Panoptic dataset. We downscale our input image by factors of 2, 4 and 8, and then upscale it back to our working resolution of $480 \times 480$, to simulate increasing the distance from the camera. The results, reported in MPJPE over all frames of the Panoptic dataset, are as follows: 138.3mm (factor of 1 - no downscaling), 140.2mm (factor of 2), 151.7mm (factor of 4) and 210.5mm (factor of 8). As expected, higher resolution of the input image correlates with better pose estimates. Critical degradation is observed only for very small people occupying a few pixels (factor of 8).

In figure 2, we show visual reconstructions of **REMIPS** on in-the-wild photos from the **COCO** validation set. Note the reconstruction quality even in high occlusion cases (either partial views or occlusion by the environment) and the robustness to the size of the humans in the image, ranging from very far away to very close to the camera. Additionally, in figure 3 we show reconstruction results on samples from the test split of **CHI3D**, with and without using the collision and contact losses. It is to be noted the better physical plausibility when using all losses.

## 5 Conclusion

To better reconstruct and analyse scenes with multiple interacting people, we have emphasized the importance of attention modeling and the role of weak supervision for scalability and good generalisation. We have introduced **REMIPS**, a new transformer-based neural network architecture for multiple person 3D pose and shape estimation from RGB images, which relies on GHUM [6], a generative and articulated full body statistical 3D model. Our network produces physically plausible human mesh reconstructions with realistic scene placement under a fully perspective camera projection model. We also introduce new collision losses uniformly applicable to both self-contacts and inter-person contact, and more efficient than previously proposed models by an order of magnitude. We demonstrate state-of-the-art 3d human pose and shape reconstruction results on challenging datasets like CHI3D[3] or Panoptic [33] which contain multiple people involved in complex interactions.

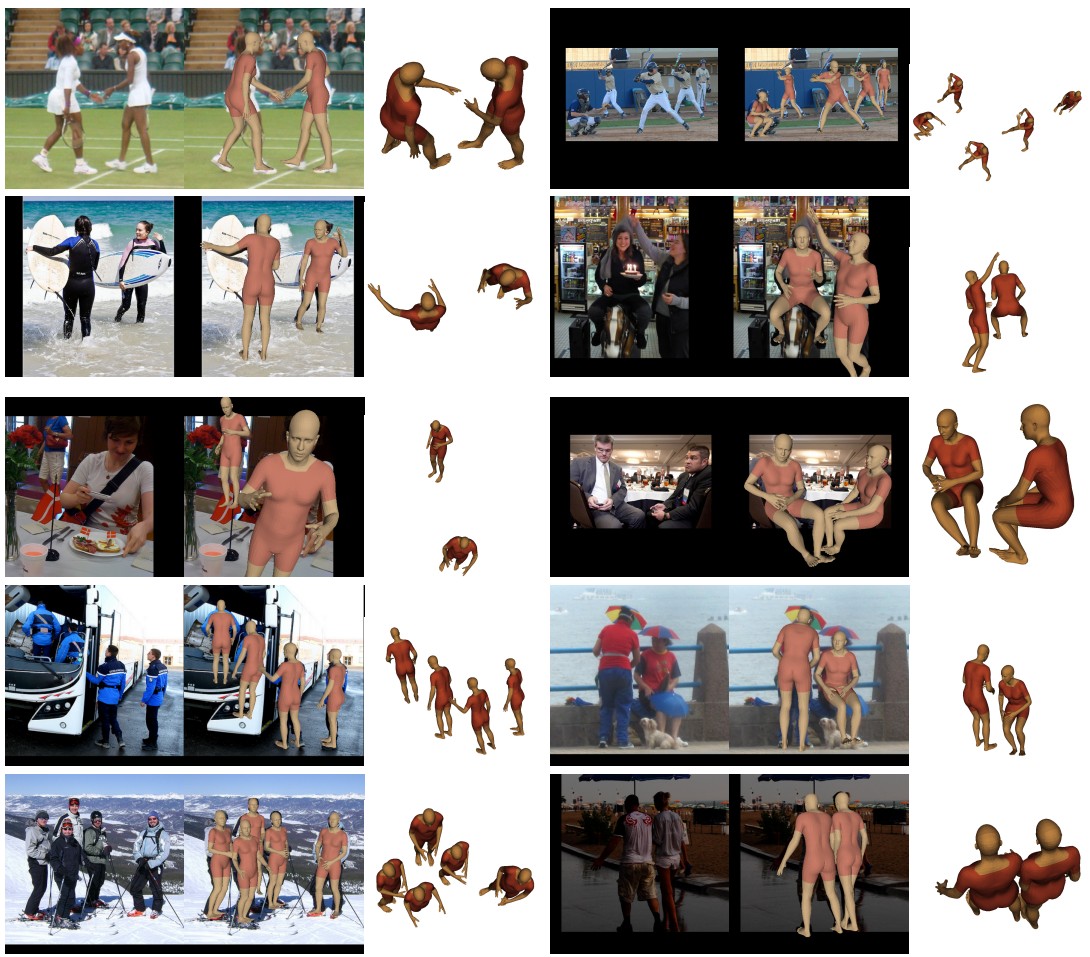

Figure 2: 3D human pose and shape predictions on the **COCO** validation set (rows 1-5) for in-the-wild images. We show the initial image together with an overlaid reconstruction of the meshes as well as a rendering from a different viewpoint which better illustrates the physical consistency of the **REMIPS** reconstructions. Best seen in color.

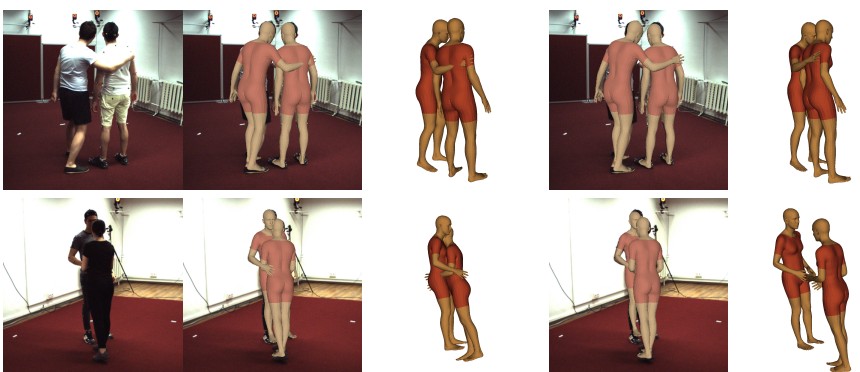

Figure 3: Visual 3D human pose and shape predictions on the **CHI3D** datasets. On columns 2-3 we show the results of a model which is not using any of the 4 losses $\mathcal{L}_{ic}, \mathcal{L}_{sc}, \mathcal{L}_{ics}, \mathcal{L}_{scs}$, while on columns 4-5 we illustrate the predictions of our full **REMIPS** model, trained using the previously mentioned 4 losses. We observe more physically plausible results when training with all losses. Best seen in color.

## Acknowledgments and Disclosure of Funding

This work was supported in part by the ERC Consolidator grant SEED, CNCS-UEFISCDI (PN-III-P4-ID-PCCF-2016-0180) and SSF.

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
