# OpenReview forum: "REMIPS: Physically Consistent 3D Reconstruction of Multiple Interacting People under Weak Supervision"
_NeurIPS.cc/2021/Conference — NeurIPS 2021 Poster_

### Official Review · Reviewer_rGCr · 2021-07-10

**Rating:** 6
**Confidence:** 4

**Summary:**

This paper presents a network and learning scheme for reconstructing multiple people from monocular images. The proposed approach includes a transformer to iteratively refine the reconstruction results, self-collision and interpenetration-collision losses based on the generalized winding numbers, and self-contact and interaction-contact losses. Qualitative results are impressive and strong.

**Limitations And Societal Impact:**

In Line 114, does the $N_P = 16$ mean that the maximum number of people in an image is 16? Do the number of people in the input image affect the performance of the transformer? Moreover, how do the solution and scale of the person in the image affect the reconstruction?

**Main Review:**

This submission basically includes incremental upgrades of existing solutions for single/multiple person reconstruction and mesh collision detection, though these solutions bring performance gains over previous approaches for multiple person reconstruction.

Methodologically, though using the transformer to leverage spatial features and update the parameters of the reconstruction is somewhat interesting, the transformer for mesh reconstruction has been investigated in [Lin et al.], which is not discussed in the paper.

[Lin et al.] End-to-End Human Pose and Mesh Reconstruction with Transformers

For collision detection, exploiting generalized winding numbers is appreciated, which provides a more efficient solution for self-collision and interpenetration penalization in multiple person reconstruction. However, it only affects the runtime during training. There is no ablation study or discussion on the performance of different solutions for collision detection.

Other minor issues:

Section 3.3 introduces the network architecture of using a transformer to predict the reconstruction results. Adding an illustration figure for the transformation of different features should make this part more readable.

There are some typos in the manuscript:
L35: deffer -> defer
L38: "so their 3D spatial relations are plausible, and the alignment with the image evidence is good.”?

**Time Spent Reviewing:**

6

---

> ### Author Response · Authors · 2021-08-10
> **Author Answer to Reviewer rGCr**
>
> **rGCr.** We are happy to reference *[Lin et al.]* in the final version of the paper. Their model reconstructs a single person at a time and is trained in a fully supervised regime and also leveraging pseudo ground truth SMPL fits for in the wild data. In contrast our model is designed for multiple interacting people and does not rely on direct 3d supervision.
>
> **rGCr. Performance of different solutions for collision detection.** Other self-collision detection solutions used in prior work [18, 19, 20, 21] rely on the set of colliding triangles and the conic distance fields they generate. This has been introduced in [19] for the task of hand pose fitting and later applied to the task of 3d full body pose optimization [20, 21] without any further ablation. As pointed out in Section 3.2.4 from [19] the choice of penalizing collisions only for colliding triangles (at the border) is **done solely** for performance reasons. Ideally, one would use the full set of points inside the mesh and their distance to the surface, which is our case. [2] does not use a self-collision loss, but only an inter-penetration loss. We show better results than [2] in Tables 1 and 2. In [3] (Section 3.3) the authors employ a self-collision loss based on a set of bounding sphere primitives. We show better results than [3] in Table 2 for the CHI3D [3] dataset. To the best of our knowledge our method is the first to propose a self-collision loss that is fast enough to be used in the context of learning a 3d pose and shape estimation network that scales to multiple people. Its usefulness is shown in the ablation from Table 3. Through our implementation of our self-collision and inter-penetration losses, we strived to achieve a common representation, easily translated across deep learning interfaces and statistical body models. We will try to ablate the other losses from the literature, with the caveat that they are slow to run and, although we did manage to port them to Tensorflow and make them applicable to the GHUM model, we did not yet manage to produce meaningful results for this rebuttal.
>
> **rGCr. The maximum number of people in an image is 16?** This number was chosen specifically for training purposes, to batch training examples together. It was extremely rare for training images to have more than 16 persons. At inference, we can actually use as many persons as person detector outputs, as long as the quadratic memory requirements for the attention mechanism are satisfied (i.e. in the order of thousands of persons). We observe in practice that the differences in inference speed for a variable number of persons (between 1 and 32) to be almost negligible,  i.e. at most 10% increase in running time.
>
> **rGCr. How does the number of people in the input image affect the performance of the transformer?** We performed further experiments on the Panoptic [32] dataset, where we ran our method in three different ways and introduced a new baseline based on SPIN[14]. The MPJPE computed under the same protocol as in [16, 1, 2] is reported, averaged across all frames and actions:
> 1) Multi-person setting - the input image is a **crop around all detected persons** and **bounding boxes are provided to the transformer together** for all persons in the crop (i.e. original version, as in the paper) - MPJPE = 138.3mm
> 2) Single-person setting with context - where the input image is **a crop around all detected persons** and **each bounding box is provided separately** to the transformer (i.e. one bounding box token at a time) - MPJPE = 138.2mm
> 3)Single-person setting without context -  where each input image consists of **a crop of only one detected person** and the transformer received only **its bounding box** (i.e. like in a standard single person 3d pose reconstruction setting, such as in SPIN [14], HMR [13], HUND [5], etc) - MPJPE = 141.7mm
> 4) Baseline - Single-person setting without context with SPIN[14] - where each input image consists of **a crop of only one detected person** that is processed by SPIN[14] - MPJPE = 164.9mm
>
> Comparing scenarios 1) and 2), the results are almost identical in terms of pose accuracy, leading us to suspect that attention to the image feature tokens is the most important part of the transformer architecture at test time, given a larger context. This means that spatial image features already encode information regarding the other persons in the scene, due to the way the network was trained (i.e. reasoning about all the persons at once). In the case of translation, the estimates do change, so the attention to the tokens associated to the other people is important. Hence, the short answer is that the number of people in the input image does not affect pose estimation performance, as long as the context is large/informative enough. However it does affect the spatial arrangement of the persons.
>
> Comparing scenarios 2) and 3), we see the benefit of providing the large context of the scene as input (4.5mm accuracy improvement).
>
> Comparing scenarios 3) and 4), we observe that even in a single-person scenario with crops tight around each person, our method performs much better than SPIN (a significant difference of 23.6mm). Note that neither of the two methods was trained on Panoptic, and our method has not used any 3d ground-truth mesh supervision. We observed large errors for SPIN in cases of occlusions and close people interactions.
>
> **rGCr. Moreover, how do the resolution and scale of the person in the image affect the reconstruction?** We performed an ablation study on the Panoptic dataset where we simulated the effect of increasing the distance of the persons in the scene (i.e the resolution of the person decreases). More precisely, we downscaled our input image by factors of 2, 4 and 8, and then upscaled it back to our working resolution of 480x480. The results, reported in MPJPE over all frames of the Panoptic dataset, are as follows: 138.3mm (factor of 1 / no downscaling), 140.2mm (factor of 2), 151.7mm (factor of 4) and 210.5mm (factor of 8). As expected, higher resolution of the input person image correlates with better pose estimates. Critical degradation is observed only for very small people occupying only a few pixels (as simulated by the factor of 8 downscaling).

---

> > ### Comment · Reviewer_rGCr · 2021-09-01
> > **Reply to Authors**
> >
> > Thanks for the response from authors. The answers have addressed most of my concerns. Overall speaking, this paper includes interesting ideas. To further improve the experiments, it is also recommended to perform an evaluation on COCO and report 2D keypoint localization results along with other approaches to 3D human mesh recovery.

---

### Official Review · Reviewer_mtiJ · 2021-07-16

**Rating:** 7
**Confidence:** 4

**Summary:**

This paper proposes a method for estimating multiple human pose & shape from monocular RGB input, which leverages a novel convolutional-transformer structure to optimize the detector-based human pose, shape and RT information. Also, it introduces a mesh-approx. based method for jointly formulating self-collision and interpenetration, which tackles the problem of treating two kinds of collisions in the same framework. Combined those methods, this paper tackles the problem of multiple human pose & shape estimation under challenging scenes.

**Limitations And Societal Impact:**

The first concern is that, although the pose and positions are estimated accurately, the estimated human shapes are slightly inaccurate, which may be further improved.
Another concern is that, although it is not fair to compare this work with monocular single-person pose & shape estimation methods, this paper will show more profundity by comparing with those methods and analyzing the results.


**Main Review:**

The paper shows very well innovation, reasonable technical ideas with robust results and sufficient analysis. The reference is comprehensive.
According to the demonstrated results, this paper shows robust results dealing with different challenging scenes, including multiple human strong occlusion, incomplete human body capturing, and strong interactions between humans. The reconstructed human poses and 3D position relations between different humans are visually accurate according to the input images. The comparisons show its robustness compared with previous works.
The proposed convolutional-transformer structure combines the architecture of CNN and transformers, and takes the CNN features and human-detection features to initialize the transformer inputs, which may inspire future works on studying better detection-reconstruction combination methods. The proposed mesh-approx. based collision resolving method moves a step forward for formulating 3D collision in deep learning networks.


**Time Spent Reviewing:**

5 hours

---

> ### Author Response · Authors · 2021-08-10
> **Author Answer to Reviewer mtiJ**
>
> **mtiJ. Human shape estimation accuracy.** One could improve the human shape estimation by using full 3d supervision. That is not currently available at scale and for images containing multiple people, filmed in the wild.
>
> **mtiJ. Comparison to single-person pose & shape estimation methods.** We produced a version of our model where we incorporated the H3.6M training set in our weakly supervised regime (we trained only with supervision from predicted 2d keypoints). No ground truth 3d information was used. We compared favorably with similar methods [5, 13] which report results for models trained without full 3d supervision. On H3.6M Protocol P2, we obtained **64mm MPJPE-PA** error and **96mm MPJPE** error while HUND (Weakly Supervised) [5] reported 66mm MPJPE-PA error and 102mm MPJPE error and HMR (Weakly Supervised) [13] reported 67mm MPJPE-PA error and 106mm MPJPE error.

---

> > ### Comment · Reviewer_mtiJ · 2021-09-01
> > **keep my original rating**
> >
> > After reading the author response to my review and other reviews and the corresponding responses, I will keep my rating to this paper as 7(good paper, accept). The author made clear analysis some quantitative experiments and comparison with other related works in the responses, and I hope these results will be revealed in the final version of paper. Although as other reviewers pointed out, the analysis of method novelty, also of whether performing comparing with some related works, can be clearer in the final paper, this paper still shows reasonable innovation and technical ideas, also with good results in multi-person interacting 3D recon scenario, which will motivate future works on this topic.

---

### Official Review · Reviewer_ueL1 · 2021-07-16

**Rating:** 6
**Confidence:** 4

**Summary:**

This paper presents a method to reconstruct multi-people in 3D from a single image. Different from popular approaches where a neural network pose estimator focuses on a single person given a bounding box, the proposed method estimates multiple people given multiple boxes at once, in particular considering their spatial layout and physical consistency. For that objective, the paper presents several novel aspects including a transformer framework, collision and interpenetration loss functions. Comparisons over previous approaches in the similar domain (multi-person) are performed in the public benchmarks (Panoptic[32] and CHI3d[3]) and the proposed framework outperforms the previous approaches.

Overall, the paper is in a good shape, containing several interesting components. Additional experiments on the other more popular dataset (3DPW) would be still needed.

**Limitations And Societal Impact:**

Yes. In section 6.

**Main Review:**

Strengths:
- The use of transformer architecture for multi-people 3D pose estimation purposes is interesting. I think that this is the first paper in that direction. The use of global image features with box features are convincing for me, to better incorporate the contextual information of the scene.

- The experiments on public benchmarks (Table 1 and 2) support the strength of the method over approaches in the same domain. The ablation study performed in Table 3 is also helpful to better understand the proposed models.

- This paper also puts effort into other details such as introducing self-collision loss and interpenetration loss. This is encouraging, even though additional experiments over alternative options would be necessary to show clearer advantages.

- I also like the fact that the paper is based on GHUM over the more popular human model, SMPL, particularly in terms of diversifying the framework in the 3D human pose estimation field.

Weaknesses:

- Missing references: There are recent work where transformer architectures are used in different ways. It would be helpful to include them with discussions.
Pose and Mesh Reconstruction with Transformers, Lin et al., CVPR 2021
Lifting Transformer for 3D Human Pose Estimation in Video, Li et al., 2021

- The paper ignores other popular 3D pose estimation benchmarks such as H36M and 3DPW. Even though the paper focuses on multi people scenarios, it would be still necessary to evaluate its performance in these datasets. In particular, note that the similar goal can be achieved via regression + optimization, for example, by (1) first estimating individual people per box (2) adjusting their spatial layout via an optimization. See the following paper:

Perceiving 3D Human-Object Spatial Arrangements from a Single Image in the Wild, Zhang et al., 2020

I understand the proposed method is trained without any 3D dataset (e.g., no H36m)? In this case,  I am concerned about its accuracy over the recent SOTA paper such as SPIN or VIBE.


**Time Spent Reviewing:**

3 hours

---

> ### Author Response · Authors · 2021-08-10
> **Author Answer to Reviewer ueL1**
>
> **ueL1. Comparison with recent transformer-based architectures.** We will gladly reference the mentioned papers. The first reference to *[Lin et al.]*  describes a fully supervised single person estimation network whereas *[Li et al., 2021]* proposes a temporal model that outputs 3D joints instead of full 3D meshes. Please note that our method is trained only in a weakly supervised regime and is specifically designed to handle multiple interacting people.
>
> **ueL1. Comparison on single person datasets.** Please see our answer to Reviewer mtiJ for results of a weakly supervised version of our model on H3.6M. We will include these numbers in the final version of the paper.
>
> **ueL1. Using a two stage approach (regression + optimization).** A two step approach based on regression + optimization would scale linearly in the number of detections, whereas our method has the same time complexity independent of the number of detections. For example, using any of the models (HMR [13], SPIN [14], HUND [5]) on each person detection box would require running the respective model on each detection separately. In contrast, in our case we only perform one forward pass. Adding optimization on top would increase the time complexity even more (usually in the order of seconds, which would be two orders of magnitude slower than our method).
>
> **ueL1. Comparison to SPIN, VIBE.** Please note that VIBE is a temporal model taking as input multiple sequential frames. SPIN [14] has the same network architecture as HMR [13], but uses the training data better. The corresponding multi-people methodology developed by the same authors as SPIN [14] and using its underlying architecture is introduced in [2]. We have compared against [2] in Tables 1 and 2.
> We additionally compare our network with SPIN[14] in a single-person scenario on the Panoptic dataset - please take a look at the experiments described in the answer to Reviewer rGCr - Scenario 3) and 4). There, both our network and SPIN take as input single-person crops around a detected person. We perform inference for each detected person at a time (using the same detector in both Scenarios). Our baseline obtains MPJPE = 141.7mm, while the SPIN [14] baseline achieves MPJPE = 164.9mm, showing that the transformer-based network, our novel losses and the multi-person training regime are superior even when inference is performed on single-person crops.

---

> > ### Comment · Reviewer_ueL1 · 2021-08-31
> > **Final rating: I keep my original rating 6.**
> >
> > I appreciate authors' responses. I still found this paper is interesting, with noticeably difference from the major stream of single person based work (e.g., HMR, SPIN). I still would like to encourage the authors to include other popular dataset (e.g., 3DPW) and comparison with single person based methods, which would provide smoother connection to the existing work. In particular 3DPW has multiple people in the scene, and it would be within this work's scenario.  I keep my original rating, 6.

---

### Official Review · Reviewer_y5WN · 2021-07-20

**Rating:** 4
**Confidence:** 3

**Summary:**

This paper presents a method for 3D shape and pose estimation of multiple persons from RGB images. The network utilizes numerous off-the-shelf techniques available from the literature and employs a transformer network to combine this information aided by a self-collision and an interpenetration loss.

**Limitations And Societal Impact:**

The paper has not discussed the limitations of the method adequately. The authors should discuss limitations associated with the points mentioned under "weaknesses" more comprehensively.

**Main Review:**

Strengths:

1. The paper is fairly well written.
2. The paper addresses an important and a valid problem.
3. The model seems to achieve good results in challenging conditions.

Weaknesses:

1. The paper uses a collection of state-of-the-art models, but the actual novelty of the method itself is limited.
2. The performance of the approach is directly limited by the performance of used off-the-shelf methods.
3. The method seems to consist of a lot of steps without proper explanations or motivations (at least the authors have failed to describe them). For example, Sec. 3.3. presents a lot of design choices but lacks an explanation on "why?". For each design choice given in Sec. 3.3, there are many alternatives. The authors should justify these choices either theoretically or using a thorough ablation study. Table 3 presents an ablation study, but it only shows the impact of loss components, which is not enough.
4. There are many different moving parts, hyper-parameters, and design choices associated with the presented method. This raises serious doubts about the robustness of the method against the slightest changes of hyperparameters and conditions (or different datasets). At least the authors should have discussed some guidelines or motivations (or at least the effects of) toward choosing hyper-parameters. In general, the performance of any model is sensitive to hyper-parameters and design choices. But in a model that is a collection of a lot of different components, this is a significant problem that should be clearly discussed.
5. The experiments section is not thorough. More experimental validation is necessary to conclude the usefulness of the model. If the paper presented a theoretically rigorous model, it is acceptable to weaken the experiments section. However, since the proposed model is mainly based on heuristics, the experiment section needs to highlight the strengths of the approach more.

**Time Spent Reviewing:**

4

---

> ### Author Response · Authors · 2021-08-10
> **Author Answer to Reviewer y5WN**
>
> **y5WN 1, 2.** Please note that except for the human detector which outputs a bounding box around detected persons in the input image, all other components of our proposed methodology are novel and have been designed and trained for this task. The fact that we use a transformer-like architecture does not imply that we use off-the-shelf models. Our methodology is the first to employ vision transformers for the task of multiple people 3D shape and pose estimation from monocular images. There are also other contributions for contact modeling, weakly supervised learning, etc. Given the qualitative and quantitative state of the art results, we argue that our technical contributions are strong.  This is not a theoretical paper as no human sensing paper is. This is applied science and technology: the value is in the novelty and in the adequacy of the proposed models, for the task, and in the quality of the results.
>
> **y5WN 3.** We report several ablation experiments which were not outlined in the paper:
> * *Using camera intrinsics as inputs.* We trained a version of our model where we didn’t use camera intrinsics as inputs. This resulted in a 15% decrease in performance on the CHI3D [3] dataset (139mm w/o camera intrinsics vs. 120mm w/ camera intrinsics MPJPE  and 158mm vs 134.7mm MPVPE in Table 2). Please also see [5] for a more detailed description of the advantages of using the camera intrinsics as inputs.
> * *Shared or separate weights for the transformer encoder layer.* When not sharing weights for the transformer encoder layer TL (L124), the network had convergence difficulties -- though we didn’t notice any increase in performance. We therefore opted for the option with fewer parameters.
> The use of a transformer-like architecture was also motivated in part by the fact that at inference time we can reconstruct a variable number of persons in a single forward pass. More details in the response to reviewer rGCr.
>
> **y5WN 4.** Choosing hyper-parameters. We did not do an exhaustive search for the 10+ hyper-parameter dimensional space (learning rate, batch size, loss weights, transformer architecture etc.), as model training is expensive in general. The loss weights were validated empirically on a smaller set, using optimization, and then fixed throughout the rest of the experiments. Other hyperparameters were each validated independently from a small subset of reasonable values. We did not observe our methodology to be very sensitive to the choice of hyperparameters and we did not finetune our parameters. We agree that the performance of the system could be improved with a more thorough search of the hyperparameter space if substantial computational resources exist (as any ML model would) but this was not the main focus of this paper.
>
> **y5WN 5.** Please note that there is a considerable amount of prior work in the field of single and multiple people 3D pose and shape estimation. We outlined some of it in the related work section. Our design choices are sound and novel. These have been demonstrated by direct experimental comparison against prior state of the art work.

---

### Author Response · Authors · 2021-08-10
**Global Author Answer**

We thank all reviewers for valuable feedback. We are glad the reviewers appreciate our paper as having multiple novel contributions, “addressing an important and valid problem for reconstructing multiple people from monocular images”, “quantitative results outperform previous SOTA approaches on several public benchmarks”, “qualitative results are impressive and strong in challenging conditions” and the paper is “well written”.  We are glad to integrate all suggestions and discuss the papers referenced by the reviewers in the final camera ready version. We will open source the models for further comparisons.

---

### Decision · Program_Chairs · 2021-09-27

**Decision:**

Accept (Poster)

**Comment:**

This submission received 4 diverging final ratings: 4, 6, 7, 6.
On the positive side, the reviewers mentioned importance of the problem, an overall interesting approach to the multi-person setting, strong performance and clear presentation. At the same time, some of the reviewers were skeptical about overall novelty (collection of known components), insufficient motivation of certain design choices and high overall complexity, lack of important ablations and evaluation on common datasets. Some of these concerns were addressed in the rebuttal, as acknowledged by the reviewers. The most skeptical reviewer did not engage in further discussion with the authors.
Overall, the AC agrees with the positively-inclined reviewers that the strengths of this work outweigh its weaknesses, and the paper presents an interesting approach that can potentially have an impact. The authors are highly encouraged to address the remaining concerns in the camera ready version. The final recommendation is to accept as a poster.